# Evolutionary Diversification of Host-Targeted *Bartonella* Effectors Proteins Derived from a Conserved FicTA Toxin-Antitoxin Module

**DOI:** 10.3390/microorganisms9081645

**Published:** 2021-07-31

**Authors:** Tilman Schirmer, Tjaart A. P. de Beer, Stefanie Tamegger, Alexander Harms, Nikolaus Dietz, David M. Dranow, Thomas E. Edwards, Peter J. Myler, Isabelle Phan, Christoph Dehio

**Affiliations:** 1Biozentrum, University of Basel, 4056 Basel, Switzerland; tdebeer@gmail.com (T.A.P.d.B.); stefanie.tamegger@unibas.ch (S.T.); alexander.harms@unibas.ch (A.H.); nikolaus.dietz@unibas.ch (N.D.); 2Seattle Structural Genomics Center for Infectious Disease, Seattle, WA 98109, USA; DDranow@be4.com (D.M.D.); tom.edwards@ucb.com (T.E.E.); peter.myler@seattlechildrens.org (P.J.M.); Isabelle.Phan@seattlechildrens.org (I.P.); 3Beryllium Discovery, Bainbridge Island, WA 98110, USA; 4Seattle Children’s Research, Seattle, WA 98109, USA; 5Departments of Pediatrics, Global Health and Biomedical Informatics & Health Education, University of Washington, Seattle, WA 98109, USA

**Keywords:** FicT/FicA toxin-antitoxin module, FIC domain, FIC signature loop, adenylylation, AMPylation, de-AMPylation, bacterial effector protein, *Bartonella* effector protein, Bep, OB-fold, BID domain, type IV secretion system, VirB/VirD4

## Abstract

Proteins containing a FIC domain catalyze AMPylation and other post-translational modifications (PTMs). In bacteria, they are typically part of FicTA toxin-antitoxin modules that control conserved biochemical processes such as topoisomerase activity, but they have also repeatedly diversified into host-targeted virulence factors. Among these, *Bartonella* effector proteins (Beps) comprise a particularly diverse ensemble of FIC domains that subvert various host cellular functions. However, no comprehensive comparative analysis has been performed to infer molecular mechanisms underlying the biochemical and functional diversification of FIC domains in the vast Bep family. Here, we used X-ray crystallography, structural modelling, and phylogenetic analyses to unravel the expansion and diversification of Bep repertoires that evolved in parallel in three *Bartonella* lineages from a single ancestral FicTA toxin-antitoxin module. Our analysis is based on 99 non-redundant Bep sequences and nine crystal structures. Inferred from the conservation of the FIC signature motif that comprises the catalytic histidine and residues involved in substrate binding, about half of them represent AMP transferases. A quarter of Beps show a glutamate in a strategic position in the putative substrate binding pocket that would interfere with triphosphate-nucleotide binding but may allow binding of an AMPylated target for deAMPylation or another substrate to catalyze a distinct PTM. The β-hairpin flap that registers the modifiable target segment to the active site exhibits remarkable structural variability. The corresponding sequences form few well-defined groups that may recognize distinct target proteins. The binding of Beps to promiscuous FicA antitoxins is well conserved, indicating a role of the antitoxin to inhibit enzymatic activity or to serve as a chaperone for the FIC domain before translocation of the Bep into host cells. Taken together, our analysis indicates a remarkable functional plasticity of Beps that is mostly brought about by structural changes in the substrate pocket and the target dock. These findings may guide future structure–function analyses of the highly versatile FIC domains.

## 1. Introduction

Fic proteins, which are characterized by containing a FIC (filamentation induced by cAMP) domain, form a diverse protein family and are found in all domains of life [1,2]. They are enzymes that mediate AMPylation (also known as adenylylation) and other post-translational modifications (PTMs) of proteins [1,2]. The FIC fold comprises eight conserved α-helices and encompasses a characteristic active site loop between helices α4 and α5 that forms most of the catalytic center and the substrate binding pocket (reviewed by [3,4,5]). Despite a remarkable diversity of target proteins and substrates, Fic proteins typically catalyze the nucleophilic attack of a target hydroxyl residue onto the diphosphate moiety of a nucleotide substrate, which causes the transfer of a phosphoryl-linked moiety onto the target protein. Most of them display a canonical (HPFx(D/E)GNGRxxR) signature motif that locates the active site loop and is critical for AMPylation activity [2,6,7,8]. However, several families of Fic proteins carry non-canonical FIC signature motifs, with some of them shown to mediate other PTMs such as phosphocholination or phosphorylation [9,10]. Recent reports showed for some Fic proteins that they are also able to deAMPylate AMPylated targets [11,12,13].

The enzymatic activity of AMPylating FIC domains is typically tightly controlled by active site obstruction via an inhibitory α-helix (α_inh_) that prevents productive ATP binding [1] and may be, in fact, crucial for de-AMPylation [11,12]. α_inh_ is found N-terminally or C-terminally at the FIC core in class II and III Fic proteins, respectively, or is part of a small interacting protein known as antitoxin in the case of class I Fic proteins [1]. Additionally, FIC domains generally contain a β-hairpin between helix α2 and helix α3 that is referred to as the “flap” and mediates docking of a target segment in extended conformation via β-strand augmentation [5,14,15,16]. This sequence-independent interaction ensures the productive insertion of the modifiable hydroxyl residue of the target into the FIC active site. On top of the catalytic core machinery, FIC domains typically contain one or more accessory extensions within or around this core that contribute to target and/or substrate specificity [3].

So far, most research on Fic proteins have focused on host-targeted virulence factors of various bacterial pathogens such as VopS of *Vibrio parahaemolyticus* and IbpA of *Histophilus somni* that inactivate Rho family GTPases by AMPylation to cause collapse of the actin cytoskeleton and host cell death [7,8]. Similarly, the type IV secretion (T4S) system effector AnkX of *Legionella pneumophila* manipulates membrane trafficking in host cells by the phosphocholination of Rab1 and Rab35 [17] and AvrAC of the plant pathogen *Xanthomonas campestris* inactivates two immune kinases by UMPylation [18]. Although the example of these proteins highlights the diversity of target proteins and PTMs of FIC domains, they only have secondarily evolved out of a much more abundant pool of Fic proteins that act in a genuine bacterial context [1]. Among these, we have previously shown that class I Fic proteins and their small inhibitory partner encompassing α_inh_ constitute FicTA toxin-antitoxin modules exemplified by VbhTA of *Bartonella schoenbuchensis* and *Ye*_FicTA of *Y. enterocolitica* [19]. When released from their FicA antitoxin, these FicT toxins cause bacterial growth inhibition by AMPylation and concomitant inactivation of type IIA topoisomerases (i.e., DNA gyrase and topoisomerase IV), causing a disruption of cellular DNA topology [19].

The α-proteobacterial genus *Bartonella* comprises numerous host-restricted species that share a conserved stealth infection strategy to cause long-lasting hemotropic infections in their respective reservoir hosts [20]. Within the genus, the amazing host adaptability and concomitant virtual ubiquity of two particular phylogenetic lineages (lineage 3 and lineage 4) have been linked to the acquisition of the VirB/D4 T4S system and the vast potential of its secreted effectors called *Bartonella* effector proteins (Beps) in order to manipulate host cell functions [21,22]. Interestingly, it appears that the VirB/D4 T4S system of lineage 3 (L3) and lineage 4 (L4) has been acquired independently with a single ancestral effector from a common source, followed by parallel series of duplication and functional diversification of the effector genes in both lineages [21]. Since adaptive radiations of L3 and L4 driven by biodiversification of these host-restricted bacteria to infect a wide range of mammals had only been triggered after the evolution of complex effector sets, it seems clear that the functional diversity of the effectors was a key innovation promoting host adaptability in the frame of the conserved *Bartonella* stealth infection strategy [21,23]. Of note, we have recently described a third acquisition of the VirB/D4 T4S system in the L1 species *B. ancashensis* that also resulted in the evolution of a complex Bep repertoire. However, this event has not yet resulted in radiation, which indicates a more primordial stage of evolution compared to L3 and L4 [24]. Although novel types of effectors without FIC domains arose in all three linages, a prominent fraction of prototypic Beps in L1 (12 out of 21), L3 (9 out of 12), and L4 (5 out of 10) contain FIC domains that are generally assumed to engage with target proteins in the host, although the molecular activity and biological role of these BepFIC domains have only begun to be revealed [21]. So far, it has been shown that Bep1 AMPylates Rac-subfamily GTPases [16], Bep2 AMPylates the host intermediate filament protein vimentin [25], and BepA causes the AMPylation of two unknown host proteins of ca. 40–50 kDa size [6]. Furthermore, all of these proteins display auto-modification, which is a common feature of FIC domains [2,6,25]. For the FIC domain of BepC with its non-canonical FIC signature motif, no enzymatic activity has been found although it binds to the RhoGEF GEF-H1 and activates this signaling protein by relocalization to the plasma membrane [26,27]. Apart from the FIC domain, all effectors contain at least one BID (*Bartonella* Intracellular Delivery) domain and a positively-charged C-terminus that forms a conserved secretion signal for translocation by the VirB/D4 T4S system [28]. Furthermore, the effectors harbor a conserved OB-fold between FIC and BID domains that was shown for other proteins to mediate interaction with oligonucleotides or oligosaccharides; however, its function in the context of the Beps remains unknown [2,6].

Taken together, in recent years we have learned a substantial amount about structure–function aspects of Fic proteins, including selected Beps, but no comprehensive comparative analysis had been performed so far to study possible molecular mechanisms underlying the biochemical and functional diversification of FIC domains in the vast Bep family. Here, we show that Beps represent genuine class I Fic proteins that can form a tight complex with the promiscuous BiaA antitoxin in *Bartonella*. Our comprehensive analysis of the structural evolution of BepFIC domains using X-ray crystallography, structural modeling, and sequence comparison shed light on the remarkable functional and regulatory plasticity of Beps brought about by minor structural changes of the FIC fold. Our results underline the diversification and specialization of *Bartonella* effectors regarding catalytic activities and target proteins and provide a paradigm for the diversification of a highly conserved enzymatic scaffold in the course of adaptive evolution.

## 2. Material and Methods

### 2.1. Bioinformatics

NCBI BLAST [29] was used to search against the nr database by only filtering out *Bartonella* sequences. The *B. rochalimae* Bep1 protein sequence was used as a query and only sequences containing both the FIC and BID domain were retained. An E-value cutoff of 1e^−3^ was used, which gave rise to 146 sequences from various *Bartonella* species as listed in Appendix A. Redundancy among these sequences was reduced by elimination of closely related homologues and resulted in 99 sequences listed in Appendix A. Full-length FIC-BID Beps sequences were annotated and aligned (CLUSTALW [30] routine with default parameters) and sequence logos were generated within Geneious Prime 2020.2.3 (www.geneious.com accessed on 10 March 2021). Protein structures were visualized with PyMOL (pymol.org accessed on 10 March 2021) and Dino (dino3d.org accessed on 10 March 2021).

The subgrouping of sequences according to active site sequence features and the computation of correlations were performed with an in-house Python (python.org accessed on 10 March 2021) routine. The correlation of the occurrence of specific residue types at two specific positions (co-conservation) was computed with another Python script. In short, for a given position in the multiple sequence alignment, the presence/absence of the respective residue type was coded with a binary of 1 or 0, respectively. This allowed the computation of the Pearson correlation between vectors representing the two specified positions. A cladogram was generated by running the Simple Phylogeny routine (European Bioinformatics Institute server (https://www.ebi.ac.uk/Tools/phylogeny/simple_phylogeny/ (accessed on 10 March 2021) on the flap segments extracted from the BepFIC multisequence alignment.

### 2.2. Cloning

*B. clarridgeiae* strain CIP 104772/73 full-length Bep1 (*Bcl*_Bep1, UniProtKB: E6YFW2, aa 1-558) and the FIC domains of *B. clarridgeiae* strain CIP 104772/73 Bep5 (*Bcl*_Bep5, E6YGF5, aa 14-226), *B. sp.* strain AR 15-3 Bep8 (*B15*_Bep8, E6YQQ1, aa 9-240), *B. sp.* strain AR 1-1C Bep8 (*B11C*_Bep8, E6YV77, aa 10-241), *B. tribocorum* strain CIP 105476/IBS 506 BepC (*Btr*_BepC, A9IWP7, aa 3-220), and *B. quintana* strain Toulouse BepC (*Bqu*_BepC, Q6FYV8, aa 3-220) were each cloned into the uncleavable pBG1861 vector [31] by LIC cloning [32] in order to produce constructs with an N-terminal hexahistidine tag.

The full-length biaA gene that codes for the small ORF directly upstream of bepA gene and part of the bepA gene from B. henselae (Bhe_bepA) encoding the FIC and OB domains (amino acid residues 1-296) were PCR-amplified from genomic DNA. The PCR products for biaA from B. henselae (Bhe_biaA) and the fragment of Bhe_bepA were cloned into the pRSF-Duet1 vector using NcoI/BamHI and NdeI/XhoI restricition sites, respectively. The pRSF-Duet1 vector containing Bhe_biaA and the Bhe_bepA constructs was transformed into E. coli BL21-AI (Invitrogen, Waltham, MA, USA). The constructs were expressed and purified as described in for VbhA/VbhT(FIC) [1] and concentrated in a crystallization buffer (20 mM HEPES, 150 mM NaCl, 2 mM MgCl_2_, 1 mM TCEP) to 20 mg mL^−1^ for crystallization.

### 2.3. Expression and Protein Purification

Each protein was expressed in BL21 (DE3) *E. coli* cells in 2l of auto-induction media in a LEX bioreactor (Harbinger, Markham, ON, Canada) at 20 °C for 72 h, after which the harvested cells were flash frozen in liquid nitrogen.

*Bcl_*Bep1 and *B15_*Bep8 FIC were purified as described previously [33]. The following protocol was used for all other proteins: The frozen cell pellet was thawed and re-suspended in 20 mM HEPES, pH 7.4, 300 mM NaCl, 5% glycerol, 30 mM Imidazole, 0.5% CHAPS, 10 mM MgCl_2_, 3 mM β-mercaptoethanol, 1.3 mg/mL protease inhibitor cocktail (Roche, Basel, Switzerland) and 0.05 mg/mL lysozyme. The collected cells were sonicated for 15 min and incubated with 20 µL Benzonase^®^ nuclease (EMD Chemicals, Gibbstown, NJ, USA) for 40 min at room temperature. The lysate was centrifuged and filtered through a 0.45 µm cellulose acetate filter (Corning Life Sciences, Lowell, MA, USA). Recombinant protein was purified by affinity chromatography by using a HisTrap FF 5 mL column (GE Biosciences, Piscataway, NJ, USA) equilibrated in 25 mM HEPES, pH 7.0, 300 mM NaCl, 5% Glycerol, 30 mM Imidazole, 1 mM DTT buffer and eluted with 500 mM imidazole in the same buffer. The concentrated sample was further purified by size exclusion chromatography in 20 mM HEPES, pH 7.0, 300 mM NaCl, 5% glycerol and 1 mM TCEP.

Fractions with pure protein (greater than 90% pure according to Coomassie-stained SDS-polyacrylamide gels) were pooled and concentrated, yielding 4.9 mg/mL for *Bqu*_BepC (expected MW = 26.47 kDa, observed MW = 26 kDa), 22.75 mg/mL of *Btr*_BepC (expected MW = 26.18 kDa, observed MW = 26 kDa), 26.4 mg/mL of *Bcl*_Bep1 (expected MW = 63.81 kDa, observed MW = 65 kDa), 45.3 mg/mL of *Bcl*_Bep5 (expected MW = 26.03 kDa, observed MW = 25 kDa, 42.3 mg/mL of B15_Bep8 (expected MW = 28.17 kDa, observed MW = 26 kDa), and 22.00 mg/mL B11_Bep8 (expected MW = 28.17 kDa, observed MW = 28 kDa. Aliquots of 100–200 μL pure protein samples were flash frozen in liquid nitrogen and stored at −80 °C.

### 2.4. Crystallography

Crystals for *Bqu*_BepC and *Btr_*BepC, *Bcl*_Bep1, *Bcl*_Bep5, *B15_*Bep8, and *B11C_*Bep8 were grown at 289 K by sitting drop vapor diffusion by mixing 0.4 μL of protein solution with 0.4 μL reservoir solution. *Bqu*_BepC FIC crystals were grown from a solution of 10% *w/v* PEG 4000, 20% *v/v* glycerol, 0.1 M bicine/Trizma base, pH 8.5, and 0.03 M each of NaF, NaBr, and NaI. *Bqu_*BepC FIC with ADP crystals were grown from a solution of 15% *w/v* PEG 3350 and 0.1 M sussinic acid. *Btr*_BepC FIC with AMPPNP crystals were grown from a solution of 10% *w/v* PEG 8000, 20% *v/v* ethylene glycol, 0.1 M MES/imdazole, pH 6.5, and 0.02 M each of sodium L-glutamate, DL-alanine, glycine, DL-lysine, and DL-serine. *Bcl*_Bep1 FIC were grown from a solution of 10% *w/v* PEG 20,000, 20% *v/v* PEG MME 5500, 0.1 M MOPS/HEPES-Na, pH 7.5, and 0.02 M each of sodium formate, ammonium acetate, trisodium citrate, sodium potassium L-tartrate, and sodium oxamate. *Bcl*_Bep5 FIC crystals were grown from a solution of 20% *w/v* PEG 3350, 0.1 M sodium citrate/citric acid, pH 4.0, and 0.2 M sodium citrate tribasic. *B11C*_Bep8 FIC crystals were grown from a solution of 30% (v/v) Jeffamine M-600 and 100 mM HEPES free acid/NaOH, pH 7.0. *B15*_Bep8 Fic crystals were grown from a solution of 20% *w/v* PEG 8000, 0.1 M HEPES/NaOH, pH 7.5.

Data for *Bqu*_BepC FIC, *B15*_Bep8 FIC, and *Bcl*_Bep5 FIC were collected at 100 K on a Rayonix MX-300 detector at a wavelength of 0.9786 Å on beamline 21-ID-G at the Advanced Photon Source (APS, Argonne, IL, USA). Data for *Bqu*_BepC FIC with ADP, *Bcl*_Bep1 FIC, and *B11C*_Bep8 FIC were collected at 100 K on a Rayonix MX-225 detector at a wavelength of 0.9786 Å on beamline 21-ID-F at the Advanced Photon Source (APS, Argonne, IL, USA). Data for *Btr*_BepC FIC with AMPPNP were collected on a Rigaku Saturn 944 + detector at a wavelength of 1.5418 Å with our in-house source (Rigaku FR-E + Superbright rotating anode). For all datasets, indexing and integration were carried out by using XDS and the scaling of the intensity data was accomplished with XSCALE [34]. For all structures except *Bqu_*BepC FIC with ADP, the structure was solved by using a molecular replacement with Phaser [35]. For *Bqu_*BepC FIC and *Bcl_*Bep1 FIC, the starting model was the *Bhe*_BepA FIC (PDB: 2JK8). For *BAR_*Bep8 FIC, *Bqu_*BepC FIC with ADP, and *Btr_*BepC FIC, the starting model was *Bqu_*BepC FIC (PDB: 4LU4). For *B11C_*Bep8 FIC, the starting model was *Bcl_*Bep1 FIC (PDB: 4NPS). For *Bcl_*Bep5 FIC, the starting model was *Bhe_*BepA FIC (PDB: 2VZA). For all structures, refinement model building was carried out by using either Refmac5 [36] or Phenix [37], TLS [38], and Coot [39]. All structures were quality checked by Molprobity [40].

The crystals for Bhe_BepA/BiaA were obtained at 22 °C by using the hanging-drop vapor diffusion method upon mixing 0.2 μL protein solution with 0.2 μL reservoir solution. The reservoir solution was composed of 0.2 M di-Sodium malonate 20% w/v Polyethylene glycol 3350. For data collection, crystal was frozen in liquid nitrogen with 20% v/v glycerol as cryoprotectant. Diffraction data were collected on beam-line X06SA (PXIII) of the Swiss Light Source (λ = 1.0 Å) at 100 K on a Pilatus 3M detector. Data were processed with XDS and the structure was solved by molecular replacement with Phaser by using the Bhe_BepA structure (PDB: 2JK8) as search model. Several rounds of iterative model building and refinement were performed by using Coot, Phenix, and Buster [41], respectively.

## 3. Results

### 3.1. Comparative Sequence Analysis of BepFIC Domains

Class I Fic toxins and their cognate antitoxins form the FicTA toxin-antitoxin module that is present in a large number of bacterial species belonging to diverse phyla [19]. A phylogenetic tree based on an alignment of the FIC domains of FicT toxins reveals multiple deep-branching clades (Figure 1). The FIC domains of *Bartonella sp.* (BepFICs) form a monophyletic cluster emerging from a deep-branching clade of rhizobial FicT toxins for which they display considerable sequence similarity. This finding indicates that they are all derived from a common ancestor. Consistent with the phylogenetic tree derived from genome-wide analysis [21,24], the BepFIC domains of *Bartonella* lineages three (Bep1–Bep8, and Bep10) and lineages four (BepA–BepC, BepI, and BepJ) and *B. ancashensis* of lineage one form separate sub-clades.

The general organization of FIC domain containing Bartonella effector proteins (called FIC-BID Beps) is shown in Figure 2A. The canonical FIC domain is found N-terminally extended by a well-conserved segment of irregular structure (N-ext). C-terminally, the BepFIC domain is followed by a small oligonucleotide/oligosaccharide-binding (OB)-like fold of unknown function and by a BID domain, which, together with the positively charged C-terminus, are responsible for T4S system-mediated translocation [24,28,42,43].

The BepFIC topology and a representative crystal structure (Bhe_BepA) are shown in Figure 2B and Figure 2C. As in other catalytically active Fic proteins (VbhTA, NmFic, etc. [1,44]) and shown for Bhe_BepC further down, the ATP substrate binds to a crevice formed by helices α1, α2, α4, and α6 with the α-phosphate moiety interacting with the N-terminus of α5. Suspended above the bound ATP substrate, there is an irregular β-hairpin (flap, orange) that has been shown to interact with target proteins (e.g., the switch-1 loop of small GTPases [14,16]) through β-sheet augmentation. Another β-hairpin (Bep element, shown in green) found exclusively in BepFICs precedes the flap. Finally, the FIC signature motif HxFx(D/E)GNGRxxR (shown in red) locates to the α4-α5 loop and the N-terminal end of α5.

The large set of non-redundant BepFIC sequences constituting the monophyletic Bep cluster shown in Figure 1 (99 sequences as listed in Appendix A) constitutes a valuable basis for the identification of functionally important residues and the classification of potential sub-groups. In the following section, BepFIC sequence conservation as represented by the sequence logo in Figure 3, is discussed in light of the known structural and functional roles of specific residues. The mean pairwise sequence identity of the BepFIC part is 46%, which is somewhat larger than the overall sequence identity of 37% calculated for the full-length FIC-BID sequences (for the full-length sequence logo, see Appendix A).

Conserved residues and segments are distributed, though not evenly, across the entire BepFIC sequence (Figure 3). In the following, the roles of these residues are discussed in relation to their position in the Bhe_BepA structure. The canonical FIC helices show only few well conserved residues and most of them are apolar and contributes to the hydrophobic core (e.g., residues L54, L75, F76, I119, L130, F144, F172, L174, L213, F214, and I217; Bhe_BepA residue numbering). The strictly conserved H72 of α2 is completely buried and the two imidazole nitrogens form H-bonds with the FIC signature loop and the loop following α2, thereby effectively tethering these loops together. There are two largely conserved inter-helix salt-bridges, R141-E215 and R193-D216, that join α4 with α7 and α6 with α7, respectively. The conservation of the aforementioned residues strongly suggests the conservation of the FIC-fold for all investigated Beps, which was confirmed by crystal structure analyses (see below).

The N-terminal extension preceding α1 shows a strongly conserved YxYPxxxxLKNKxGI motif with well-defined structure [6] (Figure 4). Noteworthy, the extension is also present in *Ec*FicT and VbhT (Appendix A). Residues Y19 and L27 of the N-terminal extension together with F80 and W82 of the FEWAG motif of the loop following α2 form a tightly packed hydrophobic core.

Following the FEWAG motif, there is a conserved insertion of 11 residues relative to *Ec*FicT and VbhT (Appendix A). This Bep specific segment (Bep element) is folded to a β-hairpin (Figure 2C and Figure 5B). Inward facing residues (F91, F93, and A99) and the tip (D95 and G96) are conserved; this suggests that it adopts the same relative position with respect to the domain core in all Beps. The Bep element is also present in the small group of rhizobial FicT toxins that is most closely related to the BepFIC domains, e.g., the FicT homologs of *Ochrobactrum anthropi* (UniProt: A6X7M7) and *Agrobacterium vitis* (UniProt: B9K658).

The subsequent “flap” involved in target registration is folded to an irregular and wide β-hairpin and hovers over the substrate binding site above the signature loop (Figure 2). The sequence of the flap shows surprising diversity both in residue composition and length. This will be discussed in further detail.

The sequence of the α3-α4 loop (Figure 3), which is located at the surface of the domain opposite to the flap, is well conserved (with the exception of Bep10, the sequence of which differs drastically from the other Beps for this loop and for α4). Most of the conserved residues (N135, L136, G138, and L139) play a structural role. The Fic signature motif shows a considerable degree of variation, which will be discussed further below. The α5-α6 loop is rather large and its sequence is mostly conserved for structural reasons. The loop allows the perpendicular arrangement of the antiparallel α6/α7 helix pair with α4 (Figure 2).

### 3.2. Comparative Structure Analysis of BepFIC Domains

As the basis for a detailed structure–function relationship study, Bep constructs were subjected to crystallization and resulted in eight new crystal structures of representative FICBeps with resolutions ranging between 3 and 1.6 Å (Appendix A). Figure 5 shows a side-by-side view and a superposition of these crystal structures and the recently determined Bro_Bep1 structure [16]. The comparison shows that the conformation and relative position/orientation of N-ext, the Bep element (green), and the FIC signature loop (red) are invariant. This contrasts with the large variability of the flap (orange), which may relate to the adaptation to various targets (see further down).

Most of the Bep crystallization constructs encompassed only the BepFIC domain, with the exception of Bro_Bep1 and Bhe_BepA, which included the adjoining C-terminal OB-fold. In these cases, the relative position/orientation of the OB-fold with respect to the FIC domain is similar but not identical. Noteworthily, in the constructs that do not have the OB-fold included, the C-terminal α8 is not fully folded (in 4WGJ, it is fully disordered) and show various orientations. Most likely, the proper folding of α8 requires the presence of the OB-fold.

### 3.3. Variability of FIC Signature Motif and Substrate Binding Site

Most FIC proteins hitherto studied catalyze target AMPylation by using ATP as substrates. These proteins show a typical HPFx(D/E)GNGRxxR sequence (FIC signature motif) for the α4-α5 linker and the adjoining start of α5. This allows the α-phosphate of the substrate to be accommodated in an anion-nest at the N-terminal end of α5 and allows the β-phosphate and γ-phosphate to be coordinated with the two arginyl residues of the motif (Figure 6A, see also references [1,14]). Latter interaction explains the second G (G2) in the motif since a side-chain in this position would interfere with binding. The histidyl side-chain is thought to act as a catalytic base to deprotonate the incoming hydroxyl side-chain of the target protein. Finally, the proline probably rigidifies the particular FIC-loop conformation, the phenylalanine anchors the motif to the hydrophobic core, the D/E side-chain takes part in Mg^2+^ coordination, the first G (G1) is required for steric reasons, and the asparagine side-chain stabilizes the loop by H-bonding back to the main-chain of the phenylalanine (see Figure 3A in [6]).

There are two more residues that are non-contiguous with the FIC signature motif but contribute to nucleotide binding in *Bqu*_BepC (this study; 4N67, 4WGJ) and VbhT ([1]; 3ZCB). These are (1) a phenylalanine from the end of the flap (*Bhe*_BepA residue number 113, also observed in IbpA and NmFic) that interacts perpendicularly with the adenine base of the substrate (and is probably also important for the fixation of the catalytic histidine) and (2) a serine/cysteine side-chain (*Bhe*_BepA number 198) that forms an H-bond with the ribose moiety of the nucleotide (Figure 6A). The overall sequence logo (Figure 6B (top)) for the non-contiguous sequence alignment (Appendix A) shows a strictly conserved PFxxGN motif, which suggests an invariant loop conformation for all of the BepFICs. However, for about half of the sequences, deviations from the strict FIC signature motif are found in the other positions suggesting that these proteins have acquired other function than catalyzing AMP transfer. This prompted us for a comprehensive and comparative sequence analysis.

The histidine required for the catalysis of AMP transfer is present in about 75% of the BepFIC sequences and the logo representing this sub-group (group a) is shown in the second row of Figure 6B. It reveals enrichment (green arrows) of F, R, and S/T in positions 113, 167, and 198, respectively, as expected for Fic proteins that are competent for nucleotide binding.

By further filtering for the presence of the second glycine (G2) and the two arginines (R1 and R2) of the canonical motif, 44 proteins are yielded (group b). Interestingly, this group almost invariably shows canonical residue types in positions 113 and 198 and N/D in position 156 (poised for ribose binding; see, e.g., Figure 6A). Therefore, we can classify group b as bona fide AMP transferases.

The second most common residue type in G2 position is glutamate (see overall logo in Figure 6B). The presence of a bulky residue in this position appears incompatible with triphosphate-nucleotide binding as will be discussed further below (Bep8 structures). 20 sequences show a glutamate in position G2 (group c) and the corresponding sequence logo in Figure 6B (bottom) clearly demonstrates that residue types required for ATP binding or AMP transfer are strongly under-represented. This suggests that group c comprises proteins with unknown functions, but see below for a further discussion.

In order to substantiate the above analysis, the co-conservation of ATP binding residues was quantified by simple pair-wise correlation calculations on all aligned sequences (see Materials and Methods). Indeed, there is a significant (>0.4) correlation between the residues that are involved in ATP binding in bona fide AMP transferases (Appendix A). Noteworthily, this correlation has no structural reason since the residues are exposed and do not interact with each other, but rather points to their common functional role. The strong anti-correlation between an E in the G2 position with canonical residue types in the other positions again points to a distinct function of the respective BepFICs.

For each group shown in Figure 6B, enumerations of Beps aggregated by lineages (L4, L3, or L1) and by orthologous sub-clades (A–C, I, J, and 1–10) are shown at the right side. Most sub-clades, but not B, 4, 5, 8, and 10, contain bona-fide AMP transferases (group b), with many orthologous groups possessing members that belong to group c (E in position G2). The remaining 20 sequences (not classified as b or c) belong to sub-clades A, I, 5–8, and 10.

### 3.4. Variability of Target Binding Flap

Fic proteins catalyze covalent modification of target hydroxyl side-chains. For this, the backbone of the target segment following the modifiable side-chain has to register to the N-terminal strand of the FIC flap (β-sheet augmentation, see Figure 7A). In our definition, the flap extends from the end of the Bep element (flap_start, P102 in *Bhe*_BepA) to the start of helix α3 (flap_end, G116 in *Bhe*_BepA, Figure 7B). Part of the flap residues point towards the FIC core and would, thus, not be in direct contact with a bound target protein. The phenylalanine in position flap_end-3 (F113 in *Bhe*_BepA) has been shown to interact with the catalytic histidine and the adenine moiety in AMP transferases (Figure 6A). The residues in positions flap_start+2 and flap_start+4 are also pointing inwards and they form two opposing walls of a slot into which the modifiable target side-chain is inserted as is observed in the IbpA/Cdc42 complex (residues L3668 and K3670 in Figure 7A). The latter residue is often a lysine or arginine which hovers over the α-phosphate of the AMPylated target tyrosine in IbpA/Cdc42 (Figure 7A) or of the ATP substrate in NmFic [1] and most likely has the additional role of stabilizing the transition state.

As noted before, Bep flaps show considerable sequence (Figure 3) and structure (Figure 7C) variation. Since the flap appears to be in a strategic position to contribute to target recognition (in addition to the sequence independent β-sheet augmentation), its variability may reflect distinct targets for individual Beps. Moreover, loss of function has to be considered.

In order to obtain insight into the variability and sub-grouping of the flap sequences, we extracted the corresponding segments from the global BepFic alignment and computed a cladogram. For this, the flap sequences were not re-aligned to ensure their unchanged relative position with the (conserved) adjoining elements. The cladogram (Appendix A) allows the definition of 12 branches composed of five to eight members each with pair-wise sequence identities between 86 and 25% (Appendix A). Although the sub-grouping may be ambiguous in some instances due to the short sequence length, the cladogram clearly indicates some well-defined branches (see the excerpt of the cladogram shown in Figure 8). Such segregation seems indicative of the functional diversification of the flaps, which is possibly related to their interaction with distinct targets.

Branches A, C, I, 1, and 7 (all being composed of members of only one sub-clade) show top-scoring pairwise flap sequence identities, which are larger than the overall sequence identity of the respective BepFICs (Appendix A). This suggests that these flap sequences are conserved for functional reasons. Figure 8 shows their sequence logos together with the pertinent part of the cladogram.

All members of branch 1 are predicted to catalyze AMP transfer (indicated in green in the cladogram). This has been verified for *Bro*_Bep1, which AMPylates the switch 1 loop of small GTPases [16]. Most members of branch A are putative AMP transferases as well. Although their target(s) have not yet been identified, it is demonstrated that *Bhe*_BepA catalysed AMPylation of proteins from the Hela cell lysate [6].

The A,B branch is interesting, since it shows a similar flap logo as branch A (Appendix A) suggesting the same target, but all of its members have a very distinct FIC motif with an E in G2 position (indicated in red in the cladogram). *Bro*_Bep2, which belongs to branch 2, was shown to AMPylate vimentin [25]. Its flap sequence resembles that of branch A, but shows a conserved D in the flap_start+6 position. However, this residue is predicted to point inwards and so it may not directly be responsible for target specificity.

### 3.5. Distinctive Features of BepFIC Domains

In general, the protein function is encoded in its three-dimensional structure. Thus, it is valuable to analyze the defining features of BepFIC sub-clades (as revealed by the sequence analyses) in the light of the representative crystal structures. The role of the individual residues of the canonical FIC motif in AMP transfer has been discussed in detail in the first chapter and elsewhere [15].

The structures of Bhe_BepA, Bcl_Bep1, Bqu_BepC, and Btr_BepC exhibit all features defining a FIC AMP transferase as described in the first chapter and elsewhere [15]. Indeed, the complex of Btr_BepC with AMPPNP (Figure 9A) demonstrates competent tri-phosphate nucleotide binding as has been observed for, e.g., the inhibition-relieved VbhT/VbhA(E24G) complex [15]. However, BepC proteins are special in that they have a lysine instead of a Mg^2+^-coordinating aspartate/glutamate in the FIC motif. The structure shows that this lysine (K150) is interacting directly with the α-phosphate. Thus, BepC proteins appear to require no divalent cation for substrate binding, in contrast to all FIC AMP transferases characterized so far. The structure also verifies that a cysteine (C185) can substitute the ribose binding serine (last position of the active site motif used in Figure 6B).

BepFIC domains with a glutamate (E159) in position G2 are represented by the *B15*_Bep8 and *B11C*_Bep8 structures, which are virtually identical. Although the motif is highly degenerated to PFxxGNE, the fold of the domain and, in particular, of the active loop are canonical (Figure 9B). Nevertheless, the binding of a triphosphate-nucleotide appears to be impossible due to the presence of the side-chain in position G2 (compare Figure 9A and Figure 9B). However, simple outward rotation of the glutamate side-chain (Figure 9C) allows the accommodation of an AMPylated side-chain in a constellation very similar to that of the Tyr-AMP target side-chain in the IbpA/Cdc42 complex [14]. Intriguingly, the model would allow binding of water molecule to the N-terminal end of α5 ready for a nucleophilic attack onto the phosphorous, which is in line with its phosphoester bond with the tyrosine, i.e., it is ready for deAMPylation of the modified side-chain. The strict conservation of a glutamate in G2 may then point to a role in water binding and proton abstraction, i.e., the glutamate would function as general base.

Members of the Bep5 sub-clade are characterized by the absence of the flap. Indeed, the structure of Bcl_Bep5 (Figure 5 and Figure 9D) shows that the Bep element is directly followed by α3. The structurally required residues of the FIC signature motif (PFxx(N/S)G) are conserved, explaining the canonical fold of the FIC loop, but most other residues of the FIC signature motif are distinct (Figure 9D).

### 3.6. Interaction of Beps with FicA Antitoxins

Class I FicT toxins are typically controlled by small FicA antitoxins. These proteins are comprised of three antiparallel helices which bind (mainly via hydrophobic interactions) to α1 of the FicT toxin, as has been shown for VbhT/A [1] and *Ec_*FicT/A [45]. The C-terminal end of the first anti-toxin helix (α_inh_) typically carries an (S/T)xxxE(G/N) motif with the glutamate forming a salt-bridge with the second arginine (R2) of the FIC signature motif. As a result, this arginine can no longer engage in ATP γ-phosphate binding, leaving the bound ATP substrate in a non-reactive orientation [1].

We have shown previously that all investigated Bartonellae encode one (or some even two) FicA homolog, which are called BiaA [24] and are homologous to VbhA. To obtain further insight into their mode of binding, several of the investigated Beps were co-expressed with their cognate BiaA and subjected to crystallization. Crystals were obtained and the structures solved for *Bhe*_BepA/*Bhe*_BiaA and *Bro*_Bep1/*Bro*_BiaA (Figure 4 and Figure 10). The two BiaA structures and their association modes with the BepFICs are very similar. The 3-helix up-and-down fold of the antitoxin is complemented by α1 of the toxin to form a canonical antiparallel 4-helix bundle, where α1 of the toxin is arranged anti-parallel to the antitoxin α1-helices and α3-helices. A large part of the antitoxin surface (about 1100 Å^2^, amounting to about 25% of the solvent accessible surface area) is buried and numerous interactions of all kinds are formed (see annotations below the sequences in Figure 10C). Although VbhA and *Ec_*FicA are only distantly related to BiaA antitoxins (22% and 12% sequence identity, respectively, w/r to *Bhe*_BiaA), a similar arrangement is observed in the VbhTA and *Ec_*FicTA complexes (Appendix A). In particular, the conserved serine/threonine and glutamate of the antitoxin motif are, again, forming interactions with the R2 side-chain of the toxin.

## 4. Discussion

The host-targeted Bep effector repertoires that evolved in parallel in three distinct *Bartonella* lineages represent a paradigm for diversifying evolution with the FIC domain mediating PTMs of target proteins representing one of their central functional units. In this work, we have explored the evolutionary trajectories of BepFIC domains along their structural and functional diversification from a single common ancestor and, ultimately, their exaptation for host interaction from a bacterial toxin-antitoxin module. As a heritage of this deep ancestry, all Bartonellae encoding FIC-BID Beps also encode at least one BiaA protein. As BiaAs forms stable complexes with BepFICs, which is exemplified by *Bro*_Bep1/*Bro*_BiaA [16] or *Bhe*_BepA/*Bhe*_BiaA (this work), they may—similarly to the homologous FicA antitoxins—inhibit AMPylation activity and possibly mediate de-AMPylation, but could also serve as chaperones for BepFICs before translocation into host cells via the T4S system. Since T4S is believed to require partial or full unfolding of its substrates, the FicA homolog will be stripped off from BepFIC domains upon translocation and thereby unleash the Beps’ activity in the eukaryotic target cell. The exceptions are the few Beps that harbor an N-terminally fused copy of an antitoxin sequence (Bep3/1, Bep3/2, and Bep4; see [24]) and they are, thus, representative of the class III Fic proteins according to the definition by Engel et al. [1].

Most class I Fic proteins were proposed to act as FicT toxins by AMPylating endogenous topoisomerases, an activity that is kept in check by the cognate FicA antitoxin [19]. For this reason, they exhibit a conserved ATP binding site and a conserved juxtaposed β-hairpin, which has the dual function to register the modifiable target segment to the active site and to contribute to target affinity/specificity.

This study showed that both the BepFIC overall structure and the backbone structure of the active site are very well conserved. However, the residue types of the active site and the sequence and structure of the target docking site show large variability, which is believed to reflect functional diversification. About half of the BepFIC domains display a canonical FIC signature motif indicating that they are phosphotransferases operating according to the well-studied catalytic mechanism with the histidine acting as a general base and a triphosphate-nucleotide as a substrate. Indeed, AMP transfer activity has been confirmed experimentally for some representatives (i.e., Bep1, Bep2, and BepA). For BepC, despite its canonical signature motif, no catalytic activity has yet been found, but its BepFIC domain mediates protein–protein interaction with the RhoGEF GEF-H1, thereby activating RhoA signaling [26,27]. It should be noted that the presence of the FIC signature motif does not permit inference about the kind of substrate nucleobase that is used by the enzyme. This is exemplified by the UMP transferase of AvrAC with its canonical motif [18].

About one quarter of BepFIC domains display a glutamate in the G2 position which would clearly interfere with the triphosphate-nucleotide binding mode observed in AMP transferases, i.e., with the γ-phosphate forming a salt-bridge with R2. However, binding of an AMPylated target to members of this group of BepFICs appears possible, which suggests that they might act as deAMPylases with the glutamate in G2 involved in binding and deprotonation of the hydrolytic water. DeAMPylation activity has been reported for the class II Fic protein FICD in humans [12] and the class III Fic protein *Ef*Fic from *Enterococcus faecalis* [11] and appears to require the “inhibitory” glutamate in a similar position as the glutamate in G2. It should be noted that only half of the members of this group have the histidine conserved and, therefore, would qualify for potential deAMPylases. It may be noted that the distantly related Fic protein Doc also has a side-chain (lysine) in G2 position, which would prevent canonical ATP binding. Indeed, it has been reported that ATP binds in an apparently inverted orientation, which explains the kinase as opposed to AMP transfer activity of Doc [10].

The remaining quarter of Beps are highly variable in their substrate binding pockets and, thus, may catalyze other PTMs or may lose enzymatic activity, but may still mediate specific effector functions such as those found for BepC [26,27].

The flap predicted to interact with the target displays considerable variability in length and amino acid composition among the Bep paralogs, but is conserved among orthologous Bep groups (Figure 8). None of the identified logos matches the well-conserved V(D/E)IxKxxxxF(A/C)xx motif of the flap of FicT toxins (see also [19] and subfamilies 1 and 2 as identified in reference [46]) from which the BepFIC domains emerged as a mono-phyletic group.

Very recently, it has been shown that small GTPases of the Rac-subfamily are the target for Bep1 and that two residues in the extended flap are critical for the exquisite target specificity [16]. With the use of modeling, this finding was rationalized by suggesting that they form two salt-bridges with Rac-subfamily specific residues. It will be most relevant for a deep understanding of BepFIC-target interactions to identify more of the targets and eventually determine the structures of the respective complexes. Whether BepA and BepB recognize the same target—due to their distinct active sites—but confer different actions (e.g., AMPylation and deAMPylation) remain an interesting question.

BepFIC domains generally carry a β-hairpin preceding the flap and a highly conserved N-terminal segment that is absent from other FIC domains. The function of this unique feature of the BepFIC domains is presently unknown, but may be related to the fact that these domains must partially unfold in the process of T4S system-mediated translocation.

The crucial role of Bep diversification for the host adaptability of the Bartonella lineages L3 and L4 is obvious from the fact that, in both cases, the adaptive radiation of these now ubiquitous groups of pathogens was triggered only after the full extant effector repertoires had evolved in their respective common ancestors [21,24]. Although much remains unknown about the biological functions of this diversity, including the more recent expansion of a complex Bep repertoire in *B. ancashensis* of L1 [24], it is clear that the variability of the BepFIC domains stick out of the huge pool of class I Fic proteins that act as FicT toxins by AMPylating bacterial topoisomerases and exhibiting a canonical or near-canonical FIC signature motif [19].

Since the lifestyle and infection cycle of *Bartonella* have largely cut them off from horizontal gene flow as an alternative source of pathogenicity factors, it seems plausible to assume that strong selective pressures have forced them to exploit the intrinsic functional versatility and plasticity of the FIC domain to the maximum. This is the opposite concept of what is represented by, e.g., *Salmonella* bacteria that profit from a continuous flow of innovations and fully functional upgrades for their virulence machineries in the spirit of “open source evolution” [47,48]. With a reference to popular culture, we favor the view that a lack of readily available and fully evolved virulence factors forced *Bartonella* to become the “MacGyver of bacterial pathogens”. In this model, strong selective pressures promoted the use of a couple of suitable raw materials to evolve highly effective and innovative, yet not necessarily elegant or elaborate pathogenicity factors de novo.

Our analysis revealed an unexpected functional plasticity of Beps that is brought about by minor structural changes in specific elements of the substrate pocket and the target dock. Future structure–function studies will be key for a better understanding of the remarkable functional plasticity of FIC domain proteins and how they contribute to host adaptability as a crucial feature of *Bartonella* evolution. This studies may also allow us to address the more fundamental question of how a conserved enzymatic scaffold, such as that of the Bep-ancestral FicT toxin that AMPylates bacterial topoisomerases, can functionally diversify over short evolutionary timescales as exemplified by the Beps in the process of host adaptation.

## Figures and Tables

**Figure 1 microorganisms-09-01645-f001:**
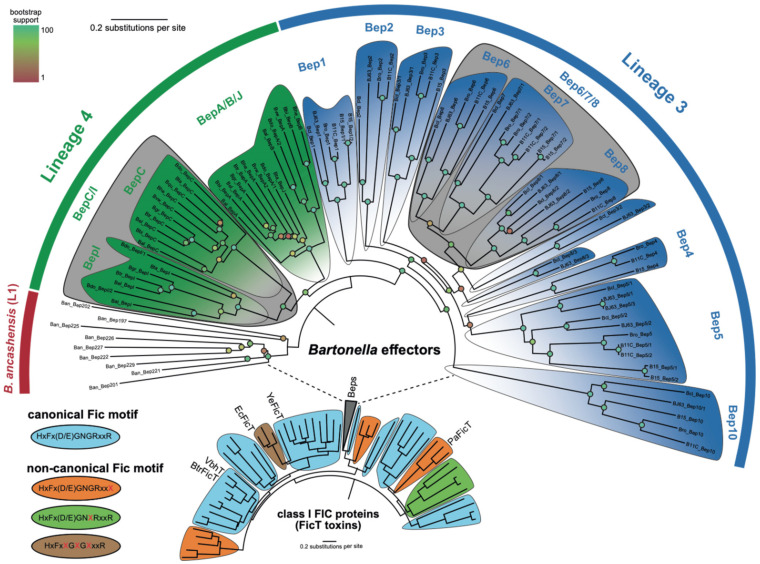
Phylogeny of class I Fic proteins. An extension of the class I Fic phylogeny published in [19], which shows the extended Bep phylogeny. The *Bartonella* Beps form part of a rhizobial clade and have ca. 35% sequence identity with them. Each clade is colored by the representative FIC motif as some Bep clades show a lot of diversity. The BepA/B clade groups together but shows a progressive change between the canonical and non-canonical FIC motif.

**Figure 2 microorganisms-09-01645-f002:**
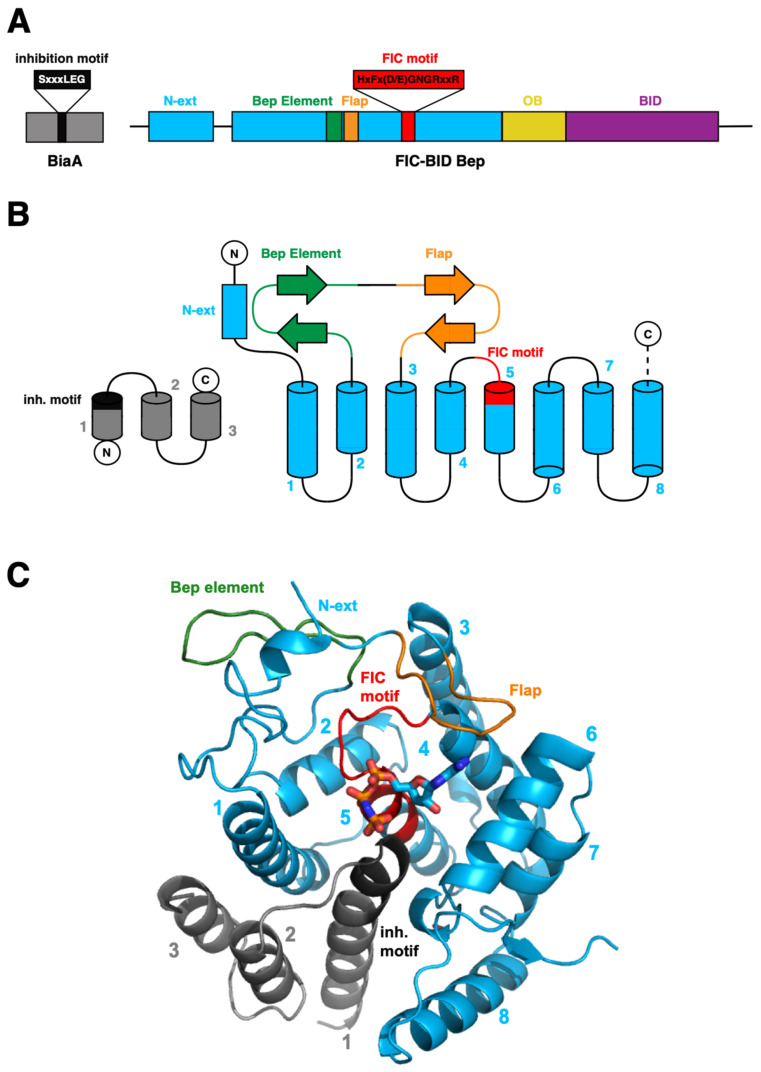
Structural organization of the BepFIC-domain and the interacting BiaA antitoxin. (**A**) Domain organization of FIC-BID proteins characterized by the FIC-fold with an N-terminal extension (N-ext) shown in blue, followed by an OB-fold domain (yellow) of unknown function and the BID-domain (magenta), which, together with the positively charged C-terminus, are responsible for T4S system-mediated secretion. (**B**) Topology of the BepFIC fold consisting of N-ext (blue), helices α1 to α8 (blue), and two β-hairpins (Bep element in green; flap in orange) between α2 and α3. The signature motif locates to the α4-α5 loop and the N-terminal end of α5 (both shown in red) that form the major part of the active site. The topology of the separate BiaA antitoxin is shown in grey. (**C**) Cartoon representation of the BepFIC domain of *Bhe*_BepA in complex with antitoxin *Bhe*_BiaA (PDB ID: 5NH2). The ATP substrate analog AMPPNP shown in sticks has been modeled based on the *Btr*_BepC/AMPPNP complex structure (4WGJ).

**Figure 3 microorganisms-09-01645-f003:**
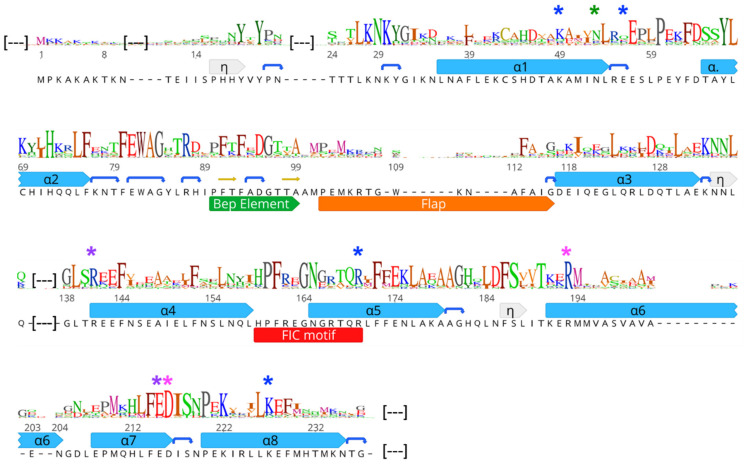
Sequence logo of BepFIC domains derived from the 99 FIC-BID Beps included in the phylogeny given in Figure 1. Sequence, numbering, and secondary structure (helices α1 to α8; 3_10_-helices; labeled η; in grey; strands in yellow; β-turns shown as blue arcs) correspond to Bhe_BepA (PDB: 5NH2). Partners of two salt-bridges are indicated by magenta and pink asterisks, respectively. Residues forming salt-bridges and the asparagine forming the Asn–Asn interaction with the antitoxin Bhe_BiaA are shown as blue and green asterisks, respectively. Functionally important segments are annotated below the sequence. Large gaps in the alignment of Bhe_BepA, due to non-representative insertions in other sequences, have been cropped for convenience and are marked by (—). The logo covering the entire length of FIC-BID Beps proteins is provided in Appendix A.

**Figure 4 microorganisms-09-01645-f004:**
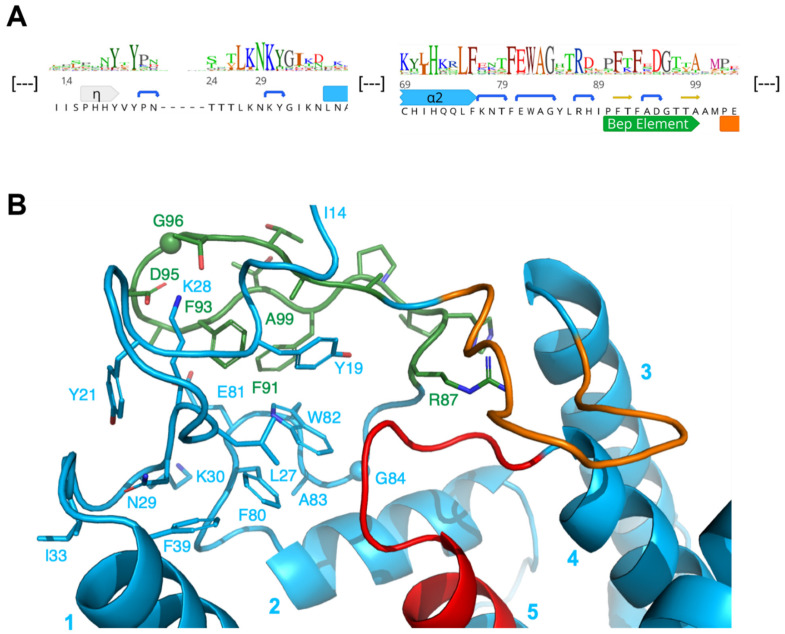
Sequence and structure of the N-terminal that is part of *Bhe_*BepA. (**A**) Excerpt of the N-terminal part of the sequence logo shown in Figure 3 with Bhe_BepA sequence below. (**B**) Detailed view of the N-terminal that is part of the Bhe_BepA structure (5NH2) showing the close interactions between the segments preceding α1 (residues 2–33), the FEWAG segment following α2 (80–84), and the Bep element (90–99, green). Various conserved residues are labeled.

**Figure 5 microorganisms-09-01645-f005:**
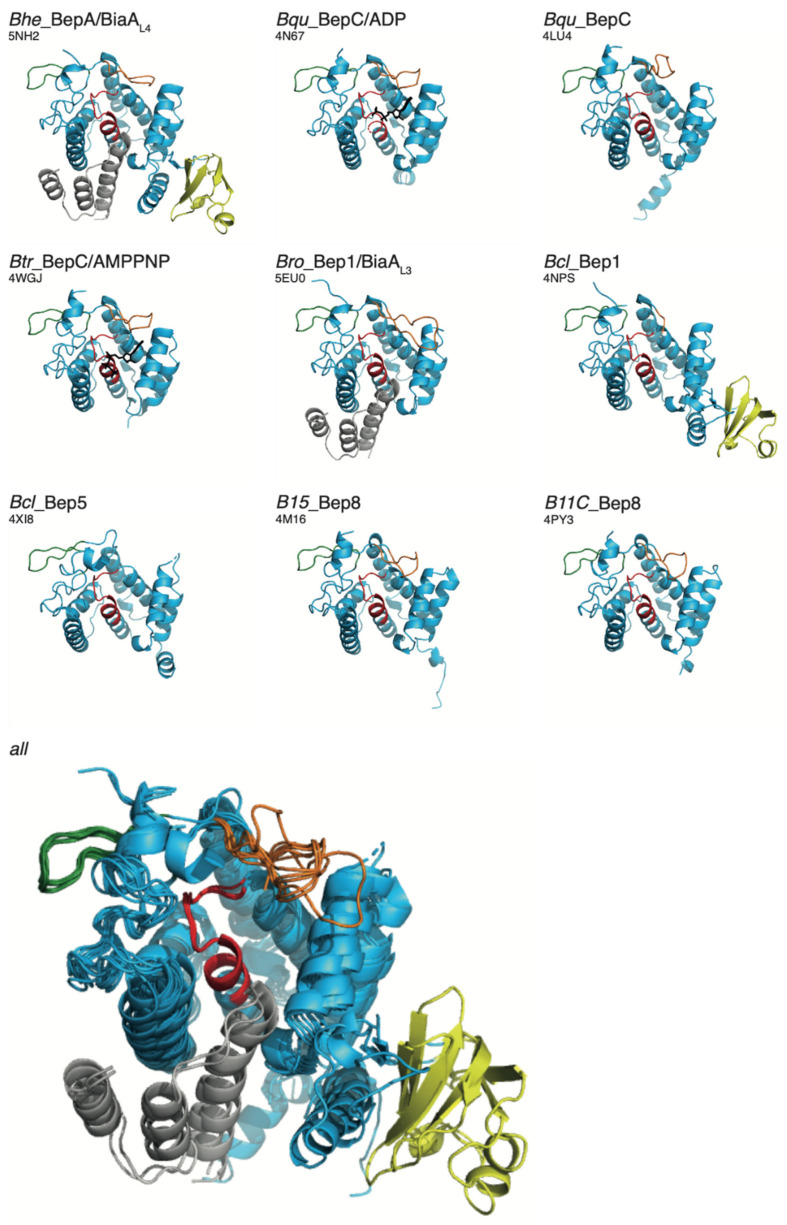
Crystal structures of BepFIC domains from various Bep sub-clades. The proteins are named according to the code provided in Appendix A and PDB codes are provided in parentheses. The structures are represented as in Figure 2. The BepFic structures of *Bhe*_BepA and *Bro*_Bep1 have been determined in complex with their cognate anti-toxins (grey) BiaA_L4_ and BiaA_L3_, respectively. The structures of *Bhe*_BepA and *Bcl*_Bep1 encompass the BepFIC and the OB-fold in yellow (see also [6]). A superposition of all structures onto *Bhe*_BepA (using α4, Fic loop, and α5) is shown at the bottom. All structures with the exception of 5EU0 [16] have been determined in this study and crystallographic details are provided in Appendix A.

**Figure 6 microorganisms-09-01645-f006:**
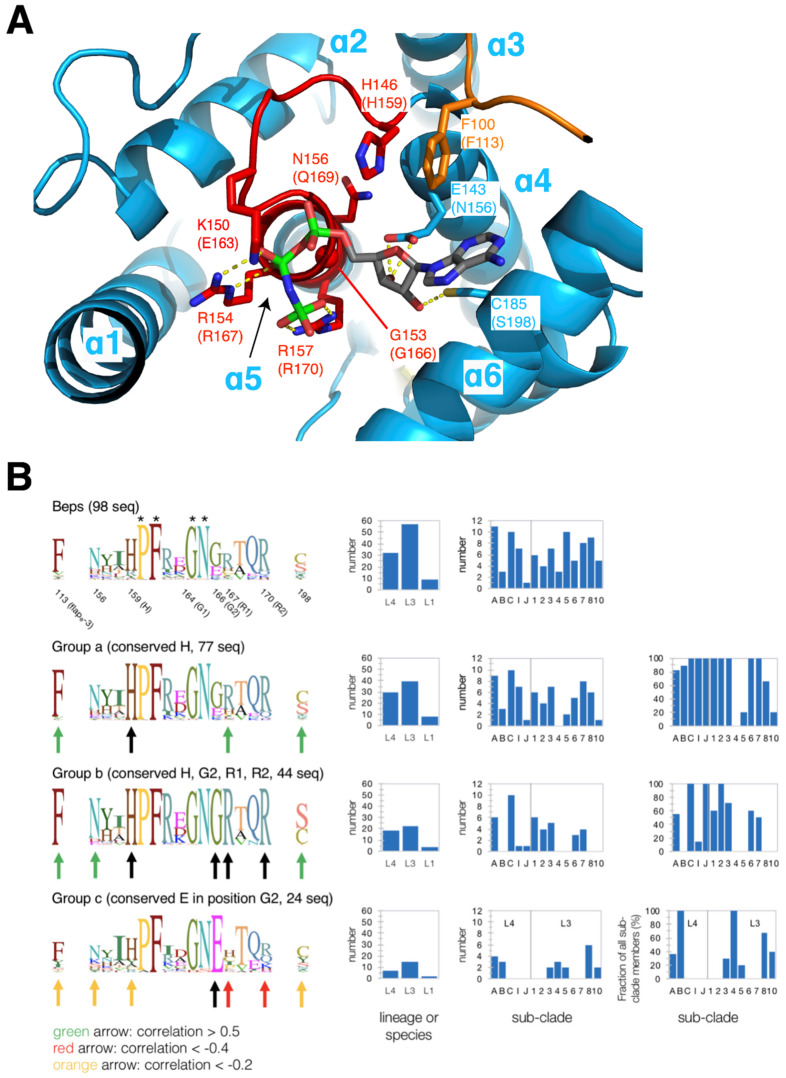
FIC substrate binding site: conservation and correlation. (**A**) Bhe_BepC structure with bound substrate analog AMPPNP (PDB ID: 4WGJ) corresponding to Figure 2C, but zoomed into the ligand binding pocket. The catalytic histidine H146 and residues interacting with the ligand are labeled (Bhe_BepA residue types and numbers in brackets). (**B**) Sequence logos encompassing the residues are shown in panel (**A**). The overall logo derived from all BepFic sequences as presented in Figure 3 is shown at the top followed by the logos of selected groups that comply to the criteria provided above the corresponding graph. Residues that show correlation or anti-correlation with the selected residue(s) (black arrows) are indicated by green (corr. > 0.5), orange (corr. < −0.2), or red (corr. < −0.4) arrows. A break-down of the group members into lineages and sub-clades (only for L3 and L4 members) is provided on the right side.

**Figure 7 microorganisms-09-01645-f007:**
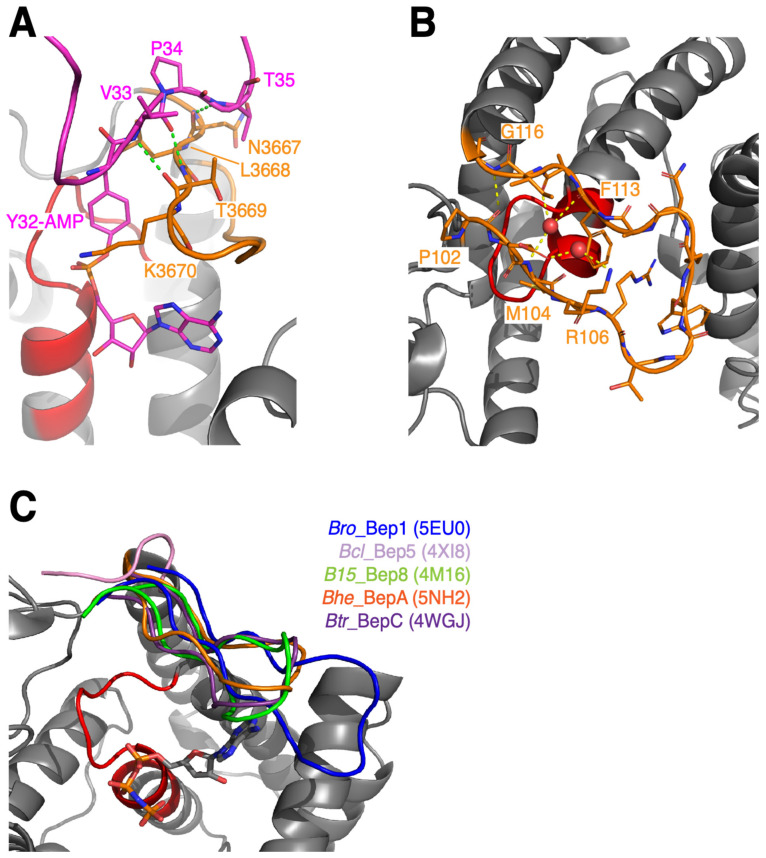
FIC flap structures as observed in the IbpA/Cdc42 complex and in BepFICs. The FIC signature motif locates to the elements shown in red and flaps are shown in orange or as indicated. (**A**) IbpA(fic2)/Cdc42-AMP (PDB: 4ITR, [14]) product complex showing registration of the target switch-1 loop with AMPylated Y32 (pink) to the N-terminal strand of the IbpA(fic2) flap via anti-parallel β-strand augmentation. (**B**) Structure of *Bhe*_BepA (PDB: 5NH2) showing the flap in detail. Note that the flap shows an open β-hairpin conformation, with some main-chain interactions (yellow dashes) mediated by water molecules (red spheres). (**C**) Comparison of BepFic flap structures (with color code as indicated) after the superposition of FIC core structures. The grey cartoon shows *Bhe*_BepA with bound AMPPNP ligand as in Figure 2C.

**Figure 8 microorganisms-09-01645-f008:**
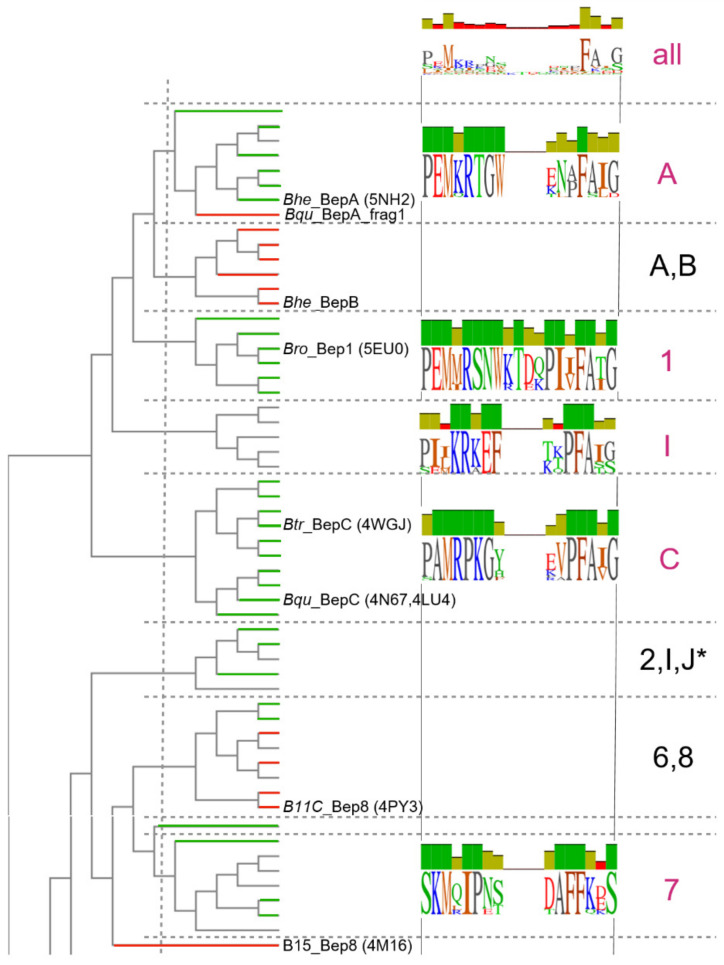
Excerpt of cladogram derived from BepFIC flap sequences with annotated sequence logos for selected branches. Horizontal lines colored in green or red indicate Beps with canonical FIC motif or with a glutamate in position G2 (groups b and d, respectively, in Figure 6B). The full cladogram with all logos and sequence names is shown in Appendix A. Branches were selected for being comprised of at least 5 members that have a pairwise identity score that is larger than the conservation score of the respective full-length sequence, see Appendix A. Branches are labeled according to the sub-clades of their members and the asterisk indicates the additional presence of Ban members (lineage 1).

**Figure 9 microorganisms-09-01645-f009:**
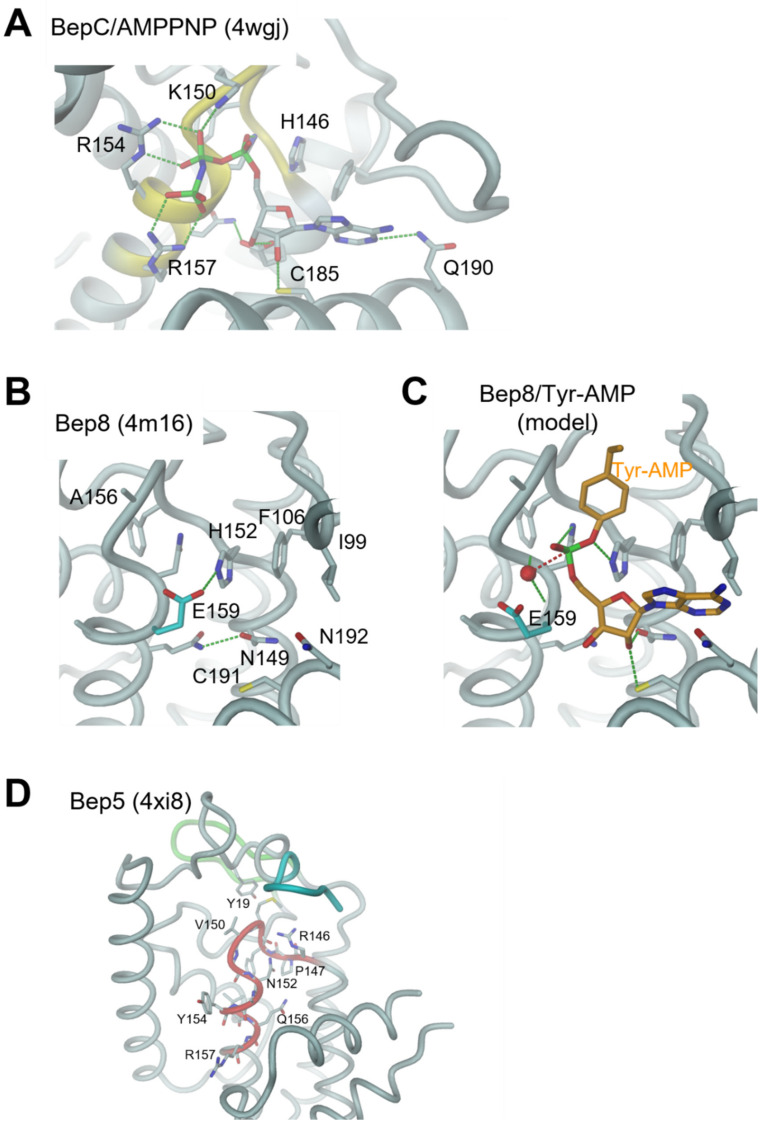
Structural details of the BepFIC domains. (**A**) *Btr_*BepC in complex with AMPPNP. (**B**) *B15_*Bep8 with E159 in position G2 highlighted in cyan. (**C**) Modeled complex of Bep8 with Tyr-AMP (orange) in a position as derived from the IbpA/Cdc42 complex 4ITR. The side-chain conformation of E159 (cyan) has been altered to allow Tyr-AMP binding. The putative hydrolytic water (red sphere) is bound to the N-terminus of α5 and possibly to E159. (**D**) *Bcl_*Bep5; note the missing flap that in other BepFIC domains is inserted between α2 and the Bep element (green). Due to disorder, K153 is shown only up to Cβ. The N-termal His-tag is indicated in cyan.

**Figure 10 microorganisms-09-01645-f010:**
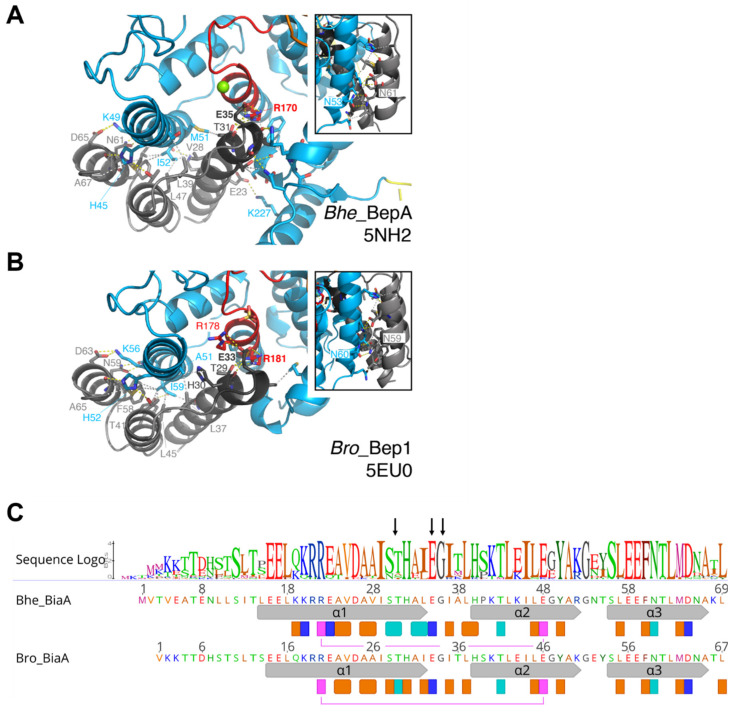
Structures of BepFIC/antitoxin complexes and antitoxin logo. (**A**,**B**) Crystal structures of *Bhe*_BepA/BiaA_L4_ (5NH2) and *Bro*_Bep1/BiaA_L3_ (5EU0); representation as in Figure 2, with interface residues shown in full. The insets show the view from the back to visualize the conserved intermolecular Asn–Asn interaction. The H-bonds between residues (cut-off 3.7 Å) are shown as yellow dashes and apolar interactions (cut-off 3.7 Å) are shown as grey dashes. (**C**) BiaA sequence logo derived from 23 *Bartonella* sequences (as defined in Figure 6 of [24], but without the divergent BiaA/2 from *B. taylori)* with annotated *Bhe*_BiaA and *Bro*_BiaA sequences underneath. Arrows above the logo indicate the residues of the inhibitory (S/T)xxxE(G/N) motif [1]. Secondary structure (grey arrows) and residues interacting with the toxin (as determined by PISA) are indicated by rectangles below the sequences (orange: apolar contact with at least 50% of the residue buried in the complex, light blue: H-bond, dark blue: salt-bridge). The magenta rectangles connected by a line indicate a conserved intra-molecular salt-bridge between α1 and α2.

## Data Availability

Crystal structures have been deposited in the PDB database (www.pdb.org) under accession codes 5NH2, 4N67, 4LU4, 4WGJ, 4NPS, 4XI8, 4M16, and 4PY3.

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
