# Peer review of "Evolutionary Diversification of Host-Targeted Bartonella Effectors Proteins Derived from a Conserved FicTA Toxin-Antitoxin Module"

_microorganisms, 2021, doi:10.3390/microorganisms9081645_

Round 1
Reviewer 1 Report
Schirmer et al., demonstrated a quality comparative analysis to investigate diversification of FIC domains in the vast Bep family. This manuscript is well designed and written. Authors have conducted crystalization assays to validate structural orientations. This manuscript needs some further modifications.
- In the introduction authors described 6 alpha helices in FIC domain, but in figure 2B its 8. Please clarify.
- Presentation of inhibitory motif in Fig 2A & B are not well integrating with FIC domain, a better presentation required.
- Include protein purification data, gel and purity yield.
Author Response
Reviewer: In the introduction authors described 6 alpha helices in FIC domain, but in figure 2B its 8. Please clarify.
Author response: In the Introduction, we refer to the 6 helices of the core fold of FIC domains. But we agree that this is confusing regarding the 8 helices shown in Fig. 2 and discussed elsewhere. So we changed
"The core fold of FIC domains comprises six conserved α-helices ...."
to
"The FIC fold comprises eight α-helices ...."
Reviewer: Presentation of inhibitory motif in Fig 2A & B are not well integrating with FIC domain, a better presentation required.
Author response: The inhibitory motif is not part of the FIC domain, but rather the BiaA antitoxin. We realize that the title of this figure does not cover the anti-toxin. So we changed
"Figure 2. Domain organization and BepFIC-domain fold. "
to
"Figure 2. Structural organization of FIC-BID Beps and BiaA antitoxins."
Reviewer: Include protein purification data, gel and purity yield.
Author response: In response to the reviewer's comment we have changed
“Fractions with pure protein (greater than 90% pure according to Coomassie-stained SDS–polyacrylamide gels) were pooled and concentrated to 20-45 mg/ml, with the exception of B. quintana BepC FIC which was concentrated to 4.9 mg/ml.”
to
“Fractions with pure protein (greater than 90% pure according to Coomassie-stained SDS–polyacrylamide gels) were pooled and concentrated, yielding 4.9 mg/ml for Bqu_BepC (expected MW= 26.47 kDa, observed MW=26 kDa), 22.75 mg/ml of Btr_BepC (expected MW= 26.18 kDa, observed MW=26 kDa), 26.4 mg/ml of Bcl_Bep1 (expected MW= 63.81 kDa, observed MW=65 kDa), 45.3 mg/ml of Bcl_Bep5 (expected MW= 26.03 kDa, observed MW=25 kDa, 42.3 mg/ml of B15_Bep8 (expected MW= 28.17 kDa, observed MW=26 kDa) and 22.00 mg/ml B11_Bep8 (expected MW= 28.17 kDa, observed MW=28 kDa.
Reviewer 2 Report
The authors have shown that the BepFIC overall structure and the backbone structure of the active site are very well conserved. Moreover, they have shown that the residues types of the active sites, the sequence and the structure of the target docking site show large viability, which is believed to reflect functional diversification.
I think the structural and the comparative sequences analysis had been well performed by the authors, although in some points the comparative is difficult to follow because they jumped from one figure to another without they indicated it in the text. For example, on page 9, when the hydrophobic core residues are numbered, it would be valued if the authors would remind that what they are describing there, it can be seen in Figure 3 and not Figure 2, which is the last cited. This would be easier to follow since it is a very descriptive article and it is continually referenced different residues.
On page 10 the same happens, figure 2 is cited but in the third paragraph figure 3 is described again and is not indicated. Moreover, in the third paragraph “the sequence of the a3- a4 loop, located at the surface of the domain is opposite to the flap, is well conserved (which the exception of Bep10…)” is not well expressed and I have to assume that the Bep10 sequence differs but the sequence is not shown.
In figure 6 I have missed several things. First of all, the 5 and 4 a helix must be indicated in the figure. And if this figure claims that to show the substrate binding pocket, it would be interested that the substrate will be docked here (As has been done in the paper that the authors cite as an example).
In page 19: Interaction of Beps with FicA antitoxin. The authors could have validated their models with some in vivo or in vitro assays with mutants in the main residues indicated as fundamental that disrupts T-AT interactions.
Minor points:
-page 20: Bartonellae vs bartonellae.
Author Response
Reviewer: Comments and Suggestions for Authors
The authors have shown that the BepFIC overall structure and the backbone structure of the active site are very well conserved. Moreover, they have shown that the residues types of the active sites, the sequence and the structure of the target docking site show large viability, which is believed to reflect functional diversification.
I think the structural and the comparative sequences analysis had been well performed by the authors, although in some points the comparative is difficult to follow because they jumped from one figure to another without they indicated it in the text. For example, on page 9, when the hydrophobic core residues are numbered, it would be valued if the authors would remind that what they are describing there, it can be seen in Figure 3 and not Figure 2, which is the last cited. This would be easier to follow since it is a very descriptive article and it is continually referenced different residues.
Response: We agree that the reference to Fig. 2C is confusing. So we changed
"Conserved residues and segments are distributed, though not evenly, across the entire length of the BepFIC domain (Fig. 3). In the following, the roles of these residues are discussed in relation to their position in the Bhe_BepA structure (Fig. 2C). "
to
"Conserved residues and segments are distributed, though not evenly, across the entire BepFIC sequence (Fig. 3). In the following, the roles of these residues are discussed in relation to their position in the Bhe_BepA structure."
Reviewer: On page 10 the same happens, figure 2 is cited but in the third paragraph figure 3 is described again and is not indicated.
Author response: We changed
"The sequence of the α3 - α4 loop, ...."
to
"The sequence of the α3 - α4 loop (Fig. 3), ...."
Reviewer: Moreover, in the third paragraph “the sequence of the a3- a4 loop, located at the surface of the domain is opposite to the flap, is well conserved (which the exception of Bep10…)” is not well expressed and I have to assume that the Bep10 sequence differs but the sequence is not shown.
Author response: The referee is right that we don't give any further information about Bep10. Still we'd like to keep the marginal information given in the brackets, which may be of interest for researchers working on Bep10.
Reviewer: In figure 6 I have missed several things. First of all, the 5 and 4 a helix must be indicated in the figure.
Author response: All helices are labeled now in the new revised version of Fig. 6.
Reviewer: And if this figure claims that to show the substrate binding pocket, it would be interested that the substrate will be docked here (As has been done in the paper that the authors cite as an example).
Author response: In Fig. 6A, we show the experimental Btr_BepC structure in complex with the substrate analog AMPPNP determined in this study. As shown in several FIC papers the AMPPNP substrate analog binds in virtually the same way to the FIC active site as the proper ATP substrate.
Reviewer: In page 19: Interaction of Beps with FicA antitoxin. The authors could have validated their models with some in vivo or in vitro assays with mutants in the main residues indicated as fundamental that disrupts T-AT interactions.
Author response: This suggestion by the reviewer is out of the scope of a minor revision and could be performed in the short deadline set by the editors for submission of the revised manuscript.
Reviewer: Minor points:
-page 20: Bartonellae vs bartonellae.
Author response: "bartonellae" is correct.